# Re-Thinking the Mediating Role of Emotional Valence and Arousal between Personal Factors and Occupational Safety Attention Levels

**DOI:** 10.3390/ijerph18115511

**Published:** 2021-05-21

**Authors:** Jiaming Wang, Pin-Chao Liao

**Affiliations:** 1School of Economics and Management, Tongji University, Shanghai 200092, China; 1551288@tongji.edu.cn; 2Department of Construction Management, Tsinghua University, Beijing 100084, China

**Keywords:** emotional valence, mediation mechanics, hazard recognition, real time

## Abstract

Emotions strongly affect occupational safety attention and public health; however, the underlying mechanisms remain unknown. We investigated the mediation mechanisms of emotional valence and arousal on safety attention using real time data. In all, 70 Chinese workers performed 8400 trials of hazard recognition tasks according to a pre-designed experiment. Their emotional and safety attention levels were recorded based on their facial expressions and eye movements, and the mediating mechanics of emotional valence and arousal were examined through a hierarchical regression. The study results show that: (1) emotional valence and arousal significantly and positively affect safety attention; (2) risk tolerance and personality significantly affect emotional valence and arousal but do not significantly affect safety attention; and (3) emotional valence and arousal significantly mediate safety attention levels and personal factors. From a theoretical viewpoint, this study corroborates the mediating role of emotion on occupational safety attention and personal factors by highlighting valence and arousal. Practically, managers can develop more specific training methods tailored to the results that pertain to workers’ higher emotional resilience for better occupational safety performance and health.

## 1. Introduction

Attention is a psychological function accompanied by mental processes that result in the concentration of mental activities on a certain task or object [1]. In the field of occupational safety and public health research, safety attention usually refers to “the concentration of employees’ psychological activities on safety issues in their working environment” [2]. In the construction industry, many workers die from safety-related accidents worldwide every year. Taking the United States as an example, according to official statistics, the number of deaths in the construction industry nationwide in 2018 was 1038. The death rate was significantly higher than that of other industries, and there has been no obvious downward trend in recent years [3]. Falls, electric shocks, and impacts caused by violations of safety regulations are among the main causes of death [3]. The theory of human errors suggests that insufficient attention to employee safety is a key internal factor that causes accidents involving humans [4]. Therefore, given the continuous occurrence of severe safety problems in the global construction industry, further improvements in ensuring the safety of construction workers and in their safety attention have become a major concern related to the sustainable and healthy development of the construction industry [5].

Existing occupational safety attention research can be classified in the following three aspects. First, in terms of the research perspective, safety attention has been discussed mainly concerning mining employees, vehicle drivers, athletes, sailors, and pilots [6], and studies on safety attention in the construction industry have also been introduced. Second, in terms of the influencing factors, much research has been conducted to elucidate organizational- and environmental-level factors that affect safety attention, such as organizational safety climate, safety commitment, and work pressure; managerial leadership; and supervision intensity, to name a few [7]. The lack of focus on individual-level influencing factors, such as emotion, risk tolerance, and personality, prevents managers from using effective measures to improve their employees’ safety performance from individual employees’ perspective. Third, in terms of the research methods, scholars have mainly adopted self-reporting methods, such as questionnaires, to measure different variables, including safety attention and emotions [8]; however, it is difficult to avoid the biases associated with these methods, which may yield inaccurate research conclusions. Thus, enrichment of individual-level influencing factors of safety attention and adoption of novel methods for measuring safety attention is needed for engineering safety management research.

Studies have shown that emotion, as an important individual-level factor, significantly affects employees’ safety and health performance [9]. The existing literature mostly discusses safety issues using concepts such as emotional quotient, emotional regulation, and emotional exhaustion. Other potentially important concepts, such as emotional valence and arousal, have not been adequately discussed. Moreover, the effect of emotion on safety performance remains controversial in academic circles. Some scholars have shown that employees with positive emotions or are excited may exhibit worse safety performance [10]. According to the theory of risk psychology, humans become more optimistic when in an excited state or swept along by positive emotions; their risk perception becomes relatively low, which likely contributes to reduced safety attention. This opinion has been reflected in the “affective generalization hypothesis” put forward by Johnson and Tversky [10]. However, many scholars have also put forward the “mood (emotion) maintenance hypothesis” [11] and pointed out that humans with positive emotions or in an excited state are often less willing to take risks because risks may negatively affect their current state of pleasure. Thus, according to this theory, humans in an excited state may become resistant to changes and are likely to pay more attention to safety hazards. In psychology, emotional positivity is usually expressed by emotional valence, and individual emotional excitement is usually expressed by arousal. Therefore, in the present work, we addressed the hypothesis that emotional valence and arousal significantly affect safety attention.

Behavioral psychology theory posits that human attention is modulated by emotion and personal characteristics, including personality and risk tolerance [12]. Since safety attention also belongs to the field of behavioral psychology, we tested the hypothesis that personality and risk tolerance affect safety attention.

To summarize, in the present study, we ask the following questions: (1) What impact do emotional valence and arousal have on construction workers’ safety attention? (2) What is the relationship between emotional valence, arousal, personality, risk tolerance, and safety attention? What is the specific mechanism of action? (3) From the perspective of emotion and personal factors, how can project managers improve their work?

Therefore, we examined the specific relationship between emotional valence, arousal, personality, risk tolerance, and safety attention. The main steps were as follows: first, the literature was reviewed, and research gaps were identified. Then, we proposed a model in which emotional valence and arousal mediated safety attention levels and personal factors. Next, to test our research hypotheses, we conducted an experiment (8400 trials of hazard recognition) on a cohort of 70 employees in a construction project in Beijing, China. Finally, based on the research conclusions, we could formulate an in-depth discussion addressing both theoretical and practical perspectives.

This study’s contributions are as follows: (1) This study proposes novel answers to the controversial issue of the impact of emotion on occupational safety performance by highlighting the effects of valence and arousal. (2) Advanced research technologies were introduced to improve the accuracy of the results. Real-time monitoring methods and artificial intelligence (AI) technology were used to measure emotional valence, arousal, and safety attention, which reduced the errors and biases associated with traditional questionnaire-based surveys. (3) Starting from individual-level factors, such as emotional valence and arousal, we provide a novel perspective for improving occupational safety attention and enriching the factors that positively affect occupational safety attention. (4) Compared with the existing research, in which hazard recognition is the only explained variable, adding safety attention as another explained variable introduces much more operability to practices, improving safety performance. Thus, we propose a novel decision-making model and develop a novel theory in the field of safety attention. (5) Practically, this paper provides novel ideas for improving construction workers’ occupational safety attention and public health and is likely to help engineering project management practices.

## 2. Literature Review and Hypotheses

### 2.1. Emotional Valence, Arousal, and Safety Attention

At present, there is a fierce controversy in academic circles regarding the relationship between emotion and safety attention. On one hand, scholars, such as Mcvay posit (based on qualitative analysis) that decreasing emotional valence increases the probability of distraction, which means that the longer the working time, the lower the emotional valence and the lower the level of concentration [13]. Li et al. analyzed samples collected from coal mines and showed that emotional exhaustion significantly and negatively affected safety attention [14]. According to the emotional maintenance hypothesis, when humans have higher emotional valence and arousal, their emotions are more positive and excited, leading to feelings of pleasure and satisfaction [15]. Feeling highly satisfied, humans increase their safety attention because any risk will negatively affect their existing state of pleasure; thus, consciously and/or unconsciously, they become more risk-averse and more change-resistant. Therefore, emotional valence and arousal may positively affect safety.

On the other hand, according to risk psychology theory, when humans are in a positive emotional state and very excited, they tend to become more optimistic, developing a relatively lower risk perception [16]. Because of the two basic characteristics of directivity and concentration [17], low risk perception leads employees to become irrational, lowering their directivity and concentration; consequently, it becomes more difficult to concentrate on safety issues in time. Therefore, emotional valence and arousal may negatively affect attention to safety.

Based on this, we propose the following two complementary hypotheses:

**Hypothesis** **1a** **(H1a).***Emotional valence and arousal positively affect the safety attention of construction workers significantly*.

**Hypothesis** **1b** **(H1b).***Emotional valence and arousal negatively affect the safety attention of construction workers significantly*.

### 2.2. Personality, Risk Tolerance, and Safety Attention

Researchers have widely adopted the Big Five personality theory to express the different personality facets—openness, conscientiousness, neuroticism, extraversion, and pleasantness [18]. In existing studies, personality has been proven to be related to employees’ safety behavior and safety-related decision-making. Nicholson et al. found that these five personality traits significantly affected employees’ risk perception, which in turn affected their safety-related decisions [19]. Through empirical analysis, Anic et al. proved that employees with a higher sense of responsibility tended to choose lower risk-sharing and, thus, exhibited higher safety performance [20]. Based on the theory of human error, Xing et al. used qualitative analysis methods to propose that personality could explain the attenuation of safety attention [21]. According to the Heinrich model, human shortcomings caused by genetic and social environments are the main factors causing accidents. These shortcomings include not only the lack of safety knowledge and skills but also defects in innate personality. Therefore, we propose that personality significantly affects the safety attention of construction workers.

As a concept originating in finance, risk tolerance has been gradually extended to the field of safety management in recent years. Risk tolerance is usually defined as “the level of risk that an individual is willing to accept when pursuing a certain goal”. Zhen et al. found that construction workers with higher risk tolerance tended to exhibit more unsafe behavior, which ultimately increased the probability of accidents [22]. A study by Ji et al. addressing pilot safety corroborated the above finding; namely, pilots with higher risk tolerance often had difficulty concentrating on safety issues, which eventually led to more unsafe operations [23]. According to the risk behavior theory, higher risk tolerance leads to lower risk perception. Based on the premise of limited energy, lower risk perception reduces human attention to risk issues, thereby reducing safety attention. Therefore, we assume that risk tolerance significantly and negatively affects safety.

In summary, we propose the following two hypotheses:

**Hypothesis** **2** **(H2).***Personality affects the safety attention of construction workers significantly*.

**Hypothesis** **3** **(H3).***Risk tolerance negatively affects the safety of construction workers significantly*.

### 2.3. Personality, Risk Tolerance, Emotional Valence, and Arousal

Based on a comprehensive analysis of the behavioral patterns of 120 construction workers, Xia proposed that risk perception significantly affects emotion [24]. Based on a sample of 297 construction workers, Wu determined that risk tolerance significantly affects construction workers’ risk perception and thus significantly affects emotion [25]. In addition, many studies in the field of psychology have shown a significant correlation between depression tolerance and emotion [26]. Since emotional valence, arousal, and risk tolerance also belong to psychology, we posit that risk tolerance can significantly affect emotional valence and arousal.

Personality is strongly correlated with emotions. In a network safety study, Norris proposed that personality differences between individuals significantly affect personal emotions and safety-related decision-making [27]. Alivernini conducted an experiment on a sample of 347 adolescents who were under social distancing restrictions during the COVID-19 pandemic and found that adolescents with different personalities exhibited significantly different emotional levels concerning social distancing restrictions and eventually developed different psychological characteristics [28]. According to the theory of emotional psychology, human psychological activities are the result of a combination of external factors (such as organizational culture) and internal factors (such as personality) [29]. Since emotion belongs to the category of mental activity, we posit that personality significantly affects emotional valence and arousal.

In summary, we propose the following two hypotheses:

**Hypothesis** **4** **(H4).***Personality affects the emotional valence and arousal of construction workers significantly*.

**Hypothesis** **5** **(H5).***Risk tolerance negatively affects the emotional valence and arousal of construction workers significantly*.

### 2.4. The Mediating Effect of Emotional Valence and Arousal

According to the discussion in Section 2.2 and Section 2.3, we hypothesize that risk tolerance and personality cannot only directly but also indirectly affect safety attention by “bridging” emotional valence and arousal. On one hand, existing studies have shown that risk tolerance and personality may significantly affect emotional valence and arousal [24,25,26,27,28,29]; on the other hand, emotional valence and arousal may also affect safety attention [18,19,20,21,22,23]. Accordingly, we propose the following hypotheses:

**Hypothesis** **6** **(H6).***Emotional valence and arousal mediate personality and safety attention*.

**Hypothesis** **7** **(H7).***Emotional valence and arousal mediate risk tolerance and safety attention*.

### 2.5. Other Hypotheses

In addition to the aforementioned seven hypotheses, there may be other correlations between emotional valence, arousal, risk tolerance, personality, and safety attention. Many studies have suggested that personality may affect risk tolerance. Sadiq studied a sample of 330 investors from Islamabad and confirmed the impact of personality traits on risk tolerance in investment decision-making [30]. Dickason and Ferraira found that investors with conservative personalities usually exhibit lower risk tolerance, which affects their investment decisions [31]. As risk tolerance is one of the several human psychological activities that can be affected by internal factors, it can be hypothesized that personality may affect it significantly.

In addition, significant mutual interactions between emotional valence and arousal have been confirmed in many studies. Guo studied athletes’ decision-making behaviors and confirmed the existence of an interaction between emotional valence and arousal using empirical analysis [32]. Natalia et al. also found an obvious correlation between the two aspects in a machine learning-based study [33]. These hypotheses differed from the above hypotheses and have been extensively validated in the literature; thus, they are brought here only for verification and are not the focus of the present study.

**Hypothesis** **8** **(H8).***Personality affects risk tolerance significantly*.

**Hypothesis** **9** **(H9).***Emotional valence positively affects arousal significantly, and arousal has a significantly positive effect on emotional valence*.

Figure 1 shows the theoretical model of the present study:

## 3. Methods

### 3.1. Data Collection

#### 3.1.1. Sample Selection

The Department of Construction Engineering Management of Tsinghua University carried out this study as an employer to study the mediating effect of emotional valence and arousal on personal factors and safety attention levels. The Infrastructure Department of Tsinghua University helped our research team contact a Chinese construction engineering company and recruited volunteers from the construction site located at Tsinghua University for this study. We then started to screen participants from the volunteers.

The screening process was divided into four steps. First, we collected basic information, including every volunteer’s age, work experience, sex, ethnicity, email address, and health status through questionnaires. In the second step, the research team selected volunteers in good health of the same or similar age (29–31), work experience, sex (male), and ethnicity (Han). In the third step, we asked the selected volunteers to participate in the experiment via email and expressed our gratitude to the unselected volunteers. Finally, to protect privacy, the research team deleted all the information of the unselected volunteers.

After repeated and careful screening, the research team selected 70 workers employed in a construction project in Beijing as the study sample. The 70 participants were all employees whom this company recruited in March 2018. They were all 29–31 years old and had no work experience related to construction engineering before joining the company, and were of almost the same age. The 70 participants were all males of Han nationality, so they belonged to the same ethnicity and sex. This sample allowed us to control for age, work experience, sex, and race, which was important given the broad socioeconomic makeup of the Chinese population. In addition, we collected the participants’ socioeconomic characteristics and conducted a descriptive statistical analysis of the participants’ basic information.

#### 3.1.2. Experiment Preparation

Before the experiment, the research team ensured three things.

First, our experiment was approved by the institutional ethics committee (project code THU201914), and we signed a legal agreement with the study participants, promising not to collect their highly sensitive information (such as mobile phone numbers, credit card information, and various passwords). As a result, we only collected each participant’s age, sex, ethnicity, area of birth, marital status, highest education level achieved, date of employment, department, work position, and workgroup. Ethics approval documents mainly included research background, research goals, methods and research materials, recruitment and randomization of subjects, research procedures, possible risks or harm to participants caused by the research, withdrawal or suspension of the research, privacy, and confidentiality.

Second, to eliminate facial feature factors that may distract from emotion recognition, the research team washed the participants’ faces to ensure that they were clean and free of beards and glasses. The research team equipped the participants with myopia with contact lenses so that the participants’ eyesight could reach the standard of normal vision without affecting the identification of potential hazards in later experiments. Therefore, artificial intelligence (AI) technology could accurately identify the participants’ emotional valence and arousal based on their facial expressions.

Third, the research team carefully designed 120 images featuring the construction site under the guidance of several professors from the Department of Construction Engineering Management and the Department of Psychology at Tsinghua University. These experts had extensive research experience in the fields of emotion, human factor engineering, and construction engineering safety. The design of 120 images was a complete process. In the first step, the research team selected 60 common safety hazards under the guidance of experts. In the second step, the research team was divided into six groups. Based on the 60 hidden hazards, we went to six construction sites and took multiple photographs related to a certain hazard at a construction site with unsafe scenes and other photographs related to the same hazard at other construction sites with safe scenes. To avoid deviations caused by memory and self-interest when workers answered questions related to the construction site where they worked, the six construction sites did not include the construction site, from which we recruited the participants. In the third step, we refined the selection of the images and selected the clearest images of the same hidden hazard (one for the safe scene and one for the unsafe scene). In the fourth step, under the guidance of experts, the research team divided each image into an area of interest (AOI) to which the study participants were expected to pay attention to determine whether the construction site was safe. In the fifth step, after the experts’ review, we also engaged three workers and three engineers in a pre-experiment. In addition to some detailed questions, the participants indicated that the questions were clear, the settings were reasonable, and the results reliable. Finally, we carefully modified the details to make them perfect. In summary, the research team meticulously designed 120 images of construction sites under the guidance of multiple experts. To identify whether there was a potential hazard in each image, the research team identified AOIs to which the study participants were supposed to pay attention. In each image, the construction site was divided into two areas: “safe” (with no potential hazard) and “dangerous” (with potential hazards).

#### 3.1.3. Experimental Process

This experiment consisted of five parts.

In the first part, a questionnaire survey was administered in the morning. The study participants answered questions regarding personality and risk tolerance. The questionnaires were completed in approximately 30 min. The experiment was conducted during the employees’ holidays so as not to delay their normal work.

The second part was a unified rest, organized at noon. The research team provided the participants with a comfortable lunch break to replenish energy, to ensure that they had sufficient physical strength to complete the test in the afternoon. After the participants rested and gathered sufficient physical strength, the hazard recognition experiment was conducted.

The third step was the hazard recognition test. In the test, 120 images were shown in random order on a computer screen. Each participant was asked to identify the potential hazard on the 120 images, choosing “safe” or “dangerous” to answer, and then 8400 repeated tests were performed. To eliminate the influence of time on emotion and to ensure that each participant’s emotions remained relatively independent when they recognized each image, the research team set a 30 s interval between responses. When the participants recognized the potential hazard, the research team used eye-tracking technology to record the length of time that each participant looked at the AOIs in real time. Simultaneously, a video recorder recorded the participants’ facial expressions and transmitted them to a computer in real time. The hazard recognition test lasted for 75–90 min.

The fourth part used AI technology to recognize facial expressions recorded in the videos. The device read emotional valence and arousal every 8 ms. The research team used the FaceReader analysis system developed by the Dutch Noldus Information Technology Company, Wageningen, The Netherlands. The system is a professional software used to analyze facial expressions automatically. It can analyze emotion-related indicators accurately and objectively, including emotional valence and arousal. The workflow’s successful operation depends on a powerful database, the AAM (Active Appearance Model) model, and the DeepFace model. As a mature analysis system, FaceReader has been used by many scholars. At present, some articles have been successfully published after utilizing the FaceReader system. For instance, the article “Applying FaceReader to Recognize Consumer Emotions in Graphic Styles” by Chia-Yin et al. used this system to finish the research [34]. Therefore, our analysis system was reliable.

The fifth part was a follow-up conducted about two weeks after the experiment through a questionnaire. There were three questions in this questionnaire: (1) What is your participant number? (2) Did you have any adverse reactions after the test? If yes, please write down specific symptoms; (3) Have you received compensation and souvenirs from us? According to the results, all participants stated that they were in a good state of mind during and after the test, and they did not feel tired or uncomfortable; the experiment did not cause any harm. Moreover, they all received compensation from our research team, including ¥100 and a souvenir from Tsinghua University.

#### 3.1.4. Ethical Statement, Data Security, and Personal Privacy

The data obtained in this study had a certain degree of sensitivity. Ensuring data security, ethics, and personal privacy are issues that are of great importance. We achieved this goal mainly through the following measures.

Before data collection, we obtained ethical approval and signed an ethical agreement with the study participants. Then, we communicated with them thoroughly, ensuring they were fully informed.

During data collection, unrelated personnel was strictly prohibited from entering and taking photos at the test site to prevent data theft. Researchers could only bring the allowed communication and filming equipment into the test site.

After data collection, we anonymized all the information and then started the analysis to protect privacy effectively. Our research team deleted all name-containing data. In addition, we did not collect highly sensitive private information, such as mobile phone numbers, credit card information, and various passwords.

After the analysis was completed, the data were stored on a safe laboratory computer’s hard drive, thus effectively preventing the data from being stolen. Moreover, we deleted all privacy-related data from the computer used for data analysis and shredded the relevant files to make them unrecoverable.

### 3.2. Measures

To conclude, the variables measured in this study mainly included emotional valence, arousal, personality, risk tolerance, hazard recognition performance, and safety attention. The following are the specific measurement methods for the variables.

#### 3.2.1. Emotional Valence and Arousal 

The research team used AI technology for video-based recognition of the participants’ emotional valence and arousal. Since both are instantaneous physical variables, this study used the participants’ average emotional valence and arousal in each trial (from starting hazard recognition to the completion of the answer) to represent the emotional valence and arousal levels.

#### 3.2.2. Personality

In this study, participants’ personality was measured using the big five personality theory, widely adopted by the research community. At present, various research institutions have developed many versions of the Big Five Personality Questionnaire (BFPQ), but many questionnaires are too complicated with redundant parts, which reduces the efficiency and accuracy of the test. Therefore, to ensure the accuracy, reliability, and simplicity of questionnaires, this research team adopted a questionnaire designed by the Department of Psychology of Tsinghua University. This questionnaire is relatively mature with good results in its application in the past 10 years. The questionnaire has 60 pictures; 12 pictures are listed in detail for each personality trait. The person with full hair in the picture is the “central figure”, who exhibits certain behavior. There are 1–2 prompt words beside each picture to describe the content of the picture. The participants were asked to assume that they were the “central figures” and then assess the possibility that they exhibited the behavior in the picture. Participants scored their possibility from 1 (0%) to 7 (100%) based on the description. After the questionnaire was completed, for every participant, we took the average score of the 12 questions as the final score of this personality trait. At present, many published studies have applied the questionnaire; for example, it was used in the article “Personality factors and safety attitudes predict safety behavior and accidents in elevator workers” published in the “International Journal of Occupational Safety and Ergonomics” [35].

#### 3.2.3. Risk Tolerance 

To measure risk tolerance, this study adopted the questionnaire designed by Ming et al. in 2011, which has been generally accepted by the academic community [23,36]. The risk tolerance questionnaire (RTQ) provides eight construction scenarios. For example, when a crane is used to lift heavy objects, the total load may exceed the threshold. Participants scored from 1 (completely unacceptable) to 5 (completely acceptable) based on their own judgment. Finally, this study considered the average score for each topic as the risk tolerance of the participants.

#### 3.2.4. Safety Attention

Following previous studies, this study considered the total length of time that the participant’s sight stayed on the AOIs in each trial to measure the level of safety attention. The variable unit was milliseconds (ms).

#### 3.2.5. Hazard Recognition Performance

If participants correctly identified the potential hazard in each image, the score for hazard recognition performance was set to 1; otherwise, it was set to 0.

### 3.3. Analytical Approach

The hierarchical linear model (HLM) is an important multivariate statistical analysis method. This method not only considers variations in the data at the same level but also variations in the data between different levels; thus, it is suitable for processing nonhomogeneous data, which endows this model with significant advantages [37]. Structural equation modeling (SEM) is another research method that has been widely used recently.

Among the five variables in this article, risk tolerance and personality were individual-related, which means each participant has only one set of data; overall, there were 70 × 1 = 70 sets of data. Emotional valence, arousal, and safety attention were trial-based; that is, each participant had 120 sets of data, and there were overall 70 × 120 = 8400 sets of data. Taking the data’s inhomogeneity and the complexity of the relationship between variables into account, this study combined the advantages of the HLM and SEM and used a two-level SEM for empirical analysis. Risk tolerance and personality are inter-individual (between individual) variables, while emotional valence, arousal, and safety attention are intraindividual (within the individual) variables.

The two-level linear SEM approach has obvious advantages [38]: (1) The advantages of both the HLM and the SEM are fully used, and the utilization of inhomogeneous data is improved; (2) the relationship between various variables in more complex models is clarified, and the reasons for the differences in safety performance between individuals are better explained; and (3) the conclusion deviations caused by the traditional linear model analysis of multilevel data are avoided.

Consequently, Mplus8.3 (Muthén & Muthén, Los Angeles, CA, USA), a statistical analysis software with the HLM analysis function, was used for hypotheses testing. Mplus8.3 has relatively simple grammatical commands, smaller statistical errors, and wider usage [39]. Moreover, this study used SPSS23.0 (IBM SPSS Statistics, Armonk, NY, USA) to perform descriptive statistical analysis and test the reliability and validity of the data.

## 4. Results

### 4.1. Measurement Model Evaluation

First, as per Section 3.1, we selected 70 volunteers to participate in our experiment and collected their basic information. Descriptive statistical analyses of this information are shown in Table 1. Teams 1–4 refer to the four parallel working groups of the project from which we recruited participants. They took turns to finish the work, and their work content was the same.

After the experiment was completed, a descriptive statistical analysis of the data for each variable was conducted. The results are shown in Table 2. The first column refers to the names of the variables, and the second one represents the levels of the variables (“within” means that the variable belongs to intraindividual level and “between” means the variable belongs to the interindividual level); *n* refers to the number of entries; the rest are the average, maximum, minimum, and standard deviation of each variable.

Third, to measure risk tolerance and personality, this study used questionnaires for data acquisition. Thus, the reliability and validity of the two variables should be tested. The specific process is as follows:

Reliability analysis of the questionnaires was conducted to determine the internal consistency of the measured results. In this study, Cronbach’s alpha (α), commonly used by researchers, was used to measure the internal consistency of the data [36]. Hair et al. pointed out that an internal consistency coefficient above 0.7 indicates that the scale used is sufficiently reliable. For exploratory research, the internal consistency coefficient can be less than 0.7 but should be above 0.6 [40].

Since risk tolerance is a single variable with α = 1, this variable is completely credible and does not require analysis. The present study used SPSS23.0 to measure the α of the personality variable because it comprises five variables. Here, α = 0.711, and we concluded that the personality variable exhibited acceptable internal consistency, indicating a high-reliability personality scale. The validity analysis of the collected questionnaires proved that the data collected by these questionnaires were suitable for this empirical study. Validity tests can determine whether scale structure classification is reasonable through factor analysis. When using factor analysis for validity testing, some prerequisites should be satisfied—there should be a strong correlation between the measured items, as reflected by the Kaiser–Meyer–Olkin (KMO) value and Bartlett’s sphericity test value. Among them, KMO, whose value is within the 0–1 range, was used to compare the simple correlation and partial correlation coefficients between the items. The criteria for this indicator are—greater than 0.9 (completely suitable), 0.7–0.9 (very suitable), 0.6–0.7 (suitable), 0.5–0.6 (not suitable), and 0.5 and below (the data should not be used) [41]. Bartlett’s sphericity test value was used to determine whether the correlation coefficient between the items was significant. If this was significant (sig. < 0.05), it was deemed suitable for the factor analysis [41].

Table 3 shows that the KMO value was 0.776, which is in the 0.7–0.9 range, indicating that the scale in this questionnaire was very suitable for further analysis. Bartlett’s sphericity test results were as follows: the chi-squared value was 223.108, which was high and proved that the corresponding *p*-value was <0.05, so Bartlett’s sphericity test was significant.

To conclude, the data’s reliability and validity met the standards for further analysis; thus, a two-level SEM evaluation was carried out.

### 4.2. Structural Model Evaluation

To test the research hypotheses, this study first used the null model to perform regression and measure the consistency and variability between groups. HLM analysis is necessary only when the differences between groups are significant; otherwise, ordinary least squares (OLS) regression can be used for analysis [42]. The null model regression equations are as follows, where *Y_ij_* is the dependent variable, $\beta_{0j}$ and $\gamma_{00}\;$are the intercepts, $\gamma_{ij}$ and $\mu_{0j}$ are residual terms, $\tau_{00}$ is the inter-individual variance and $\tau^{2}$ is the intra-individual variance.
(1)$$Y_{ij}=\beta_{0j}+\gamma_{ij}$$
(2)$$\beta_{0j}=\gamma_{00}+\mu_{0j}$$
(3)$$Variance\left(Y_{ij}\right)=\tau_{00}+\tau^{2}$$
(4)$$Variance\left(\beta_{0j}\right)=\tau_{00}$$

Internal consistency indicators were computed, including *ICC (1)*, *ICC (2)*, and *r_wg_* of safety attention, emotional valence, and arousal. The criteria were: *r_wg_* > 0.7, *ICC (1)* > 0.12, and *ICC (2)* > 0.7 [43]. The corresponding equations are shown below, where *S_a_* refers to the within-group variance, *S_b_* refers to the random variance, and *k* refers to the number of samples at the individual level.
(5)$$ICC\;\left(1\right)=\rho =\frac{\tau_{00}}{\tau_{00}+\tau^{2}}$$
(6)$$ICC\;\left(2\right)=\frac{k\cdot ICC\left(1\right)}{1+\left(k-1\right)\cdot ICC\left(1\right)},$$
(7)$$r_{wg}=1-\frac{S_{a}}{S_{b}},$$

The calculation results are listed in Table 4. The *ICC (1)* values of safety attention, emotional valence, and arousal were 0.350, 0.643, and 0.584, respectively, all above 0.12; the *ICC (2)* values were 0.974, 0.992, 0.990, all above 0.7; the ranges of *r_wg_* were, respectively, [0.73, 0.91], [0.81, 0.95], and [0.79, 0.87], all above 0.7. Therefore, the differences between the groups were obvious, indicating that it was necessary to perform an HLM analysis.

Next, we adopted a complete model regression for HLM analysis. In the equations below, *Y_ij_* denotes the explained variables. The results of this analysis are shown in Figure 2.
(8)$$Y_{ij}=\beta_{0j}+\beta_{1j}X_{ij}+\gamma_{ij}$$
(9)$$\beta_{0j}=\gamma_{00}+\gamma_{01}W_{j}+\mu_{0j}$$
(10)$$\beta_{1j}=\gamma_{10}+\gamma_{11}W_{j}+\mu_{1j}$$

After the analysis, the following conclusions were drawn:

Emotional valence and arousal significantly and positively affected safety attention; thus, H1a was verified, and correspondingly, H1b was rejected. Personality did not significantly affect the safety attention of construction workers; thus, H2 was rejected. Risk tolerance positively affected the safety attention of construction workers; thus, H3 was rejected. Personality significantly affected the emotional valence and arousal of construction workers; thus, H4 was confirmed. Risk tolerance significantly and negatively affected the construction workers’ emotional valence and arousal; thus, H5 was confirmed. Emotional valence and arousal played a mediating role between personality and safety attention; thus, H6 was confirmed. Emotional valence and arousal played a mediating role between risk tolerance and safety attention; thus, H7 was confirmed. Personality did not significantly affect risk tolerance; therefore, H8 was rejected. Emotional valence significantly and positively affected arousal, and arousal significantly and positively affected emotional valence; thus, H9 was confirmed.

Consequently, H1a, H4, H5, H6, H7, and H9 were confirmed. To illustrate the results of our empirical analysis, we show the best-fit SEM in Figure 3.

To test the practicability and reliability of the above results, this study used hazard recognition performance as the explained variable in the model instead of safety attention. After the HLM regression analysis, the results shown in Figure 4 were obtained. The study results suggest that although the significance levels are slightly different, the hypotheses were confirmed when safety attention was used as the explained variable and remained confirmed when hazard recognition performance was used as the explained variable. This not only proves that the conclusions of this study are theoretically reliable but also indicates that this study can effectively improve safety performance in practice.

## 5. Discussion

Given that the global construction industry continues to face some safety and health issues that are difficult to solve, we think that our present findings may have a certain theoretical and practical significance concerning improving occupational safety and public health.

### 5.1. Theoretical Contribution

In terms of contributing to safety attention, this study may provide a relatively novel perspective and method for occupational safety and public health research. At present, safety attention research has been introduced in fields such as mining, transportation, sports, and construction engineering. For example, Dongling et al. studied the relationship between emotion regulation ability and the safety of archers [44]; Meiting et al. conducted an empirical analysis of the safety attention distribution in a cohort of sailors [45], and Yin considered a sample of 137 miners working in the coal industry and explored the factors affecting safety attention [7]. In construction engineering, existing research on safety attention mainly focuses on the two levels of organization and environment. Zhang et al. systematically analyzed the organizational- -and environmental-level factors influencing safety attention [2], and Siu et al. studied a sample of construction workers in Hong Kong and discussed factors, such as organizational safety climate [46]. The lack of relevant research on individual-level influencing factors has confused the management practices of engineering managers and led to defective theories related to safety attention. For instance, what individual-level factors significantly affect safety attention? What individual-level factors do not? What influencing factors should managers prioritize for improving safety performance? This study used a sample of construction workers to design and perform experiments with controlled variables. The study results suggested direct positive effects of emotional valence and arousal and indirect effects of risk tolerance and personality on safety attention. This study has enriched the research on safety attention in construction engineering, elucidated the factors affecting safety attention at the individual level, clarified the relationship between safety attention levels and personal influencing factors, and unveiled the mechanisms by which various variables affect safety attention.

In terms of emotional valence and arousal, this study provides a novel perspective on the role of emotion in occupational safety. On one hand, existing research on emotional valence has been focused on physiological, medical, and imaging aspects, with little attention paid to engineering safety [47,48,49]. On the other hand, much controversy exists in academic circles regarding the relationship between emotion and safety performance. Some scholars believe that emotion positively affects safety attention; that is, they support the “mood (emotion) maintenance hypothesis” [11]. Others believe that emotions negatively affect safety attention based on risk psychology analysis [10]. In the existing research related to safety performance, scholars have mostly used the concepts of emotional quotient, emotional regulation, and emotional exhaustion to capture and quantify emotions. This article did not use these concepts but instead introduced emotional valence and arousal into the field of safety attention. The present study results strongly support the “emotion maintenance hypothesis” based on our experiments and provide a valuable reference for understanding the connection between emotion and safety performance.

To measure emotional valence, arousal, and safety attention, this study selected a novel method to improve the accuracy of measuring emotions and safety attention. The conventional metrics of emotional valence, arousal, and safety attention are typically assessed using questionnaire surveys, Likert scale methods, and other methods; all of these methods are still widely used. For instance, in the study on emotional valence, arousal, and autonomous driving published by Du et al. in 2021, scale scoring was still used to measure emotional valence and arousal [50]. When scholars, such as Sato et al. analyzed movie watchers, a similar method was used to quantify the physiological correlation between emotional valence and arousal [51]. Tardif-Grenier et al. built a safety attention model based on vulnerability theory and used a questionnaire survey method for measuring safety attention. Methods of this sort generally exhibit the following problems [52]: (1) The accuracy and completeness of the questionnaire setting are difficult to guarantee; (2) The misunderstanding and subjective answers of the respondents cannot be eliminated thoroughly; and (3) Emotional valence, arousal, and safety attention are continuous variables, but the scores of the Likert scale can only be integers, leading to inaccurate measurement results. To improve the measurement accuracy and reduce deviation, we adopted a dynamic real-time monitoring method for obtaining video recordings of the participant’s facial expressions and used AI methods for measuring emotional valence and arousal. At the same time, eye-tracking technology successfully identified the length of time the participants’ sight was focused on the AOI. All of the above methods allowed the recording of emotional valence, arousal, and safety attention more objectively and as continuous variables instead of giving integer scores by participants based on subjective judgments. Therefore, the introduction of advanced technology has improved the accuracy and objectivity of the measurements.

### 5.2. Practical Contribution

Traditional occupational safety attention research mainly focuses on organizational- and environmental-level influencing factors, so managers in the construction engineering field often find it difficult to effectively improve employees’ personal safety and health performance. The results of the present study suggest that emotional valence and arousal significantly and directly affect construction workers’ safety attention. Personality and risk tolerance did not significantly and directly affect construction workers’ safety attention but indirectly affected it by “bridging” emotional valence and arousal. This discovery, including individual-level influencing factors and mechanisms, provides useful guidance for management practices. To improve safety performance and public health based on this theoretical analysis, this study puts forward the following three suggestions for project safety management practices:(1)Concentrate on the safety attention of construction employees and give full play to the spillover effect of improving safety attention. In the conventional engineering safety management model, managers are often confused owing to the lack of entry points for improving employees’ safety performance and health. This study explored engineering safety management from the perspective of safety attention, and the study results showed that safety attention was highly correlated with hazard recognition performance. Therefore, safety attention can be used as an important entry point to improve safety performance. Because the traditional safety management model lacks concentration on safety attention, project managers and grassroots employees are expected to jointly establish awareness to improve safety attention. Safety attention training links should be added to induction training and regular employee training. In addition, the training process should not be limited to oral presentations and written reports, which means that field visits and hands-on practices can also be added to achieve better training outcomes.(2)Pay attention to the mediating role of emotional valence and arousal and use emotion as a platform for enhancing employees’ safety attention. This study confirmed the mediating effect of emotional valence and arousal concerning safety attention and personal factors. Therefore, emotional valence and arousal can be regarded as a “bridge” or “platform” for improving the employees’ safety attention. In the context of the emergence of “intelligent buildings”, “intelligent construction”, and “intelligent monitoring” on a global scale, managers can rely on advanced and intelligent technologies for “intelligent management” and “fine management”. For example, managers can test the emotional valence and arousal of workers every day before they start work, remind depressed employees to concentrate, and help workers with low emotional valence and arousal to enhance their safety awareness. In addition, we suggest that managers use various methods, such as team building, one-on-one psychological counseling, and relaxation during construction projects to improve the employees’ emotional valence and arousal. Thus, managers can prevent accidents as much as possible.(3)Comprehensively consider the indirect impact of risk tolerance and personality to help employees establish risk prevention awareness. This study showed that personality had no significant direct impact on risk tolerance and its impact on safety attention was indirect. Therefore, when recruiting workers, managers should not place too much emphasis on the personality traits of candidates or adopt discriminatory policies but should strengthen induction training and education, especially risk tolerance education for employees. Existing studies have shown that risk tolerance is affected by organizational, environmental, and individual factors at multiple levels. Therefore, managers are expected to strive to help employees reduce their risk tolerance from multiple perspectives. For example, they can actively create a corporate safety culture, strengthen safety education and training for employees, and conduct safety knowledge and skill competitions to help employees establish a risk prevention outlook.

Concerning the relevant recommendations of the Health and Safety Executive of the UK (HSE), this study proposes a refined management model based on the plan-do-check-act (PDCA) management cycle model and social norm theory [53,54]. First, before recruiting employees, we advise companies to formulate emotion-related competency standards based on the characteristics of each job position. Second, after recruitment, managers can arrange their positions according to the employees’ characteristics and carry out corresponding training. Training may focus on employees’ emotional regulation ability and environmental adaptability and should strive to reduce employees’ risk tolerance through safety education. Third, a daily inspection system can be set up for employees’ emotional valence and arousal before they start their jobs, safety reminders can be strengthened for employees with lower indicators, and appropriate psychological counseling and physiological adjustments can be provided. Finally, the system’s effectiveness based on changes in safety performance should be checked, and successful experiences and shortcomings can be summed up to continuously improve the details of safety management. For more details, please refer to Figure 5.

Construction engineering enterprises, construction workers, and managers are the main actors in the field of construction engineering. Some large Chinese companies, including the State Grid of China, are planning to explore the effect of emotions on safety management. Large enterprises are recommended to play a demonstrative and leading role in implementing smart construction, improving safety and productivity, and assuming corporate social responsibility. The trickle-down effect will eventually benefit small-and-medium-sized enterprises. For these enterprises, the fast pace of work, lower staffing levels, and greater financial pressure may induce managers to think that it is relatively difficult to implement our current recommendations; however, these enterprises can also make useful explorations at this stage. First, the equipment for recognizing emotions is not expensive and is being economized and popularized. The equipment enables the recognition of emotions very quickly, and signals can be read out in real time. Therefore, the implementation of emotion-related management models is worth the investment. Second, improving safety productivity and reducing accident rates can bring large economic benefits to enterprises, including both reduced accident compensation and the potential economic benefits offered by the improvement of corporate reputation. Third, a number of small and medium-sized enterprises can form a consortium on safety production issues, jointly invest in the purchase of relevant equipment, and establish a refined management model to reduce individual enterprises’ costs.

The government is expected to be a supervisor and a helper in the field of construction engineering. Reducing the accident rate has positive implications for governments. The public sector can strengthen the supervision of safe production, clarify the safety requirements in the field of construction engineering by legislation, and appropriately raise safety standards as technology advances. Moreover, we think the government is obliged to introduce policies that are conducive to developing small and medium-sized enterprises and support the enterprises that implement novel technologies to ensure the safety of their employees in various forms through measures, such as subsidies, preferential treatment, and tax cuts.

In summary, it is recommended that the government and enterprises of all sizes should develop benign interactive relationships to jointly act to improve the safety level of construction projects.

## 6. Conclusions

This study aimed to explore the specific relationships among emotional valence, arousal, personality, risk tolerance, and safety attention. The empirical analysis found that (1) emotional valence and arousal significantly and positively affected safety attention; (2) emotional valence and arousal play a mediating role between safety attention levels and personal factors; (3) personality and risk tolerance have no direct influence on safety attention, but significantly affect emotional valence and arousal, and (4) personality has no significant influence on risk tolerance, and the interaction between emotional valence and arousal was confirmed.

This study explored the existing controversial opinions about the relationship between emotion and safety performance, supported the “emotion maintenance hypothesis,” and provided new references for solving these controversial issues. At the same time, this study adopted a dynamic real-time monitoring method and introduced AI technology to overcome the drawbacks of conventional questionnaire surveys for measuring emotion, provided novel methods and ideas for occupational safety attention research, and built a novel safety attention theoretical model seeking to enrich the theory of safety attention. Furthermore, this study laid a solid foundation for effectively improving the safety, attention and health of construction workers at the individual level and put forward suggestions and guidance for improving the safety performance of engineering projects.

Although this study has a unique value, it also has certain limitations, which offer new avenues for future research.

First, owing to the research conditions, the sample in this study came from a certain Chinese construction site. Therefore, future researchers are expected to select employees from different countries as a study sample to validate the conclusions of this study further. Additionally, future scholars can also consider adding other construction sites from different locations in China to verify our results.

Second, although emotional valence and arousal have mediating effects, other variables may mediate the relationship between safety attention and personal factors as well. Although these additional mediating variables are beyond the scope of the present work, future studies should address this issue.

Third, in addition to individual-level factors, organizational- and environmental-level factors may also affect safety attention. Thus, future research should fully integrate the three levels (individual, organizational, and environmental) to establish more complex and more complete models to further explore safety attention mechanisms.

## Figures and Tables

**Figure 1 ijerph-18-05511-f001:**
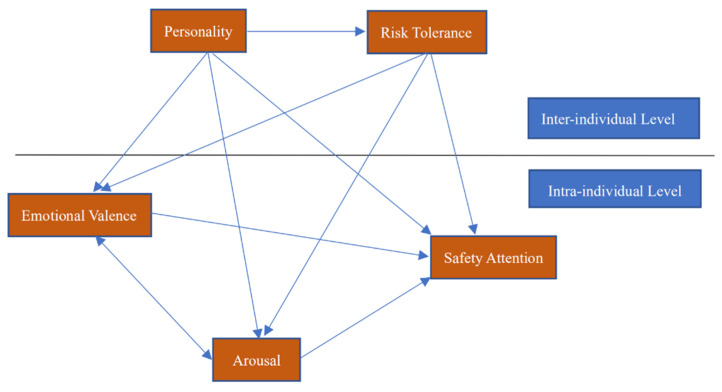
Theoretical model.

**Figure 2 ijerph-18-05511-f002:**
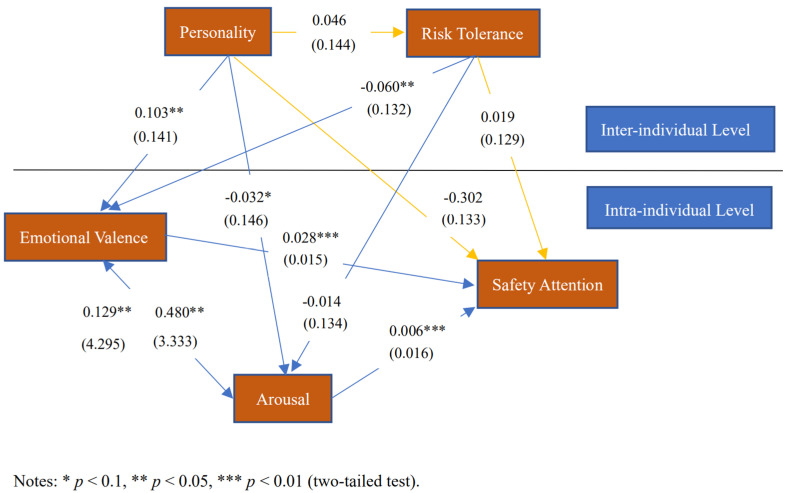
Initial test of the hypothesized structural model.

**Figure 3 ijerph-18-05511-f003:**
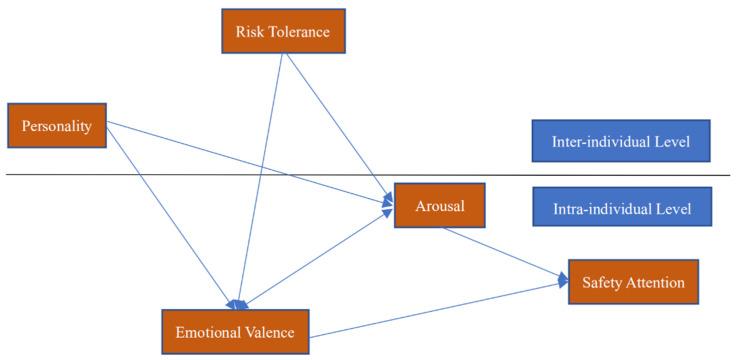
Best-fit SEM (Structural equation modeling).

**Figure 4 ijerph-18-05511-f004:**
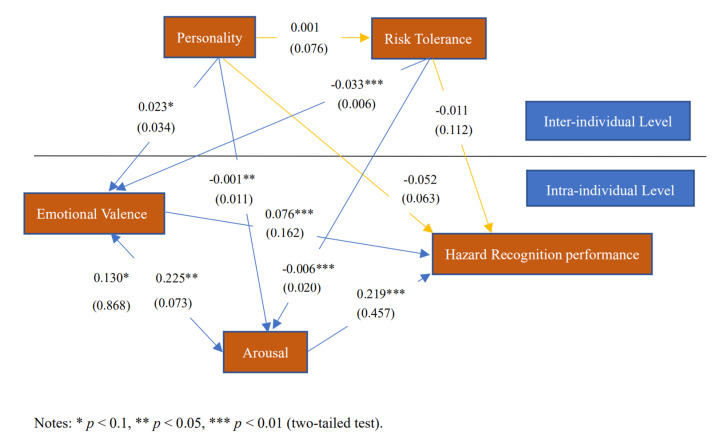
Initial test of the structural model, taking the hazard recognition performance as an explained variable.

**Figure 5 ijerph-18-05511-f005:**
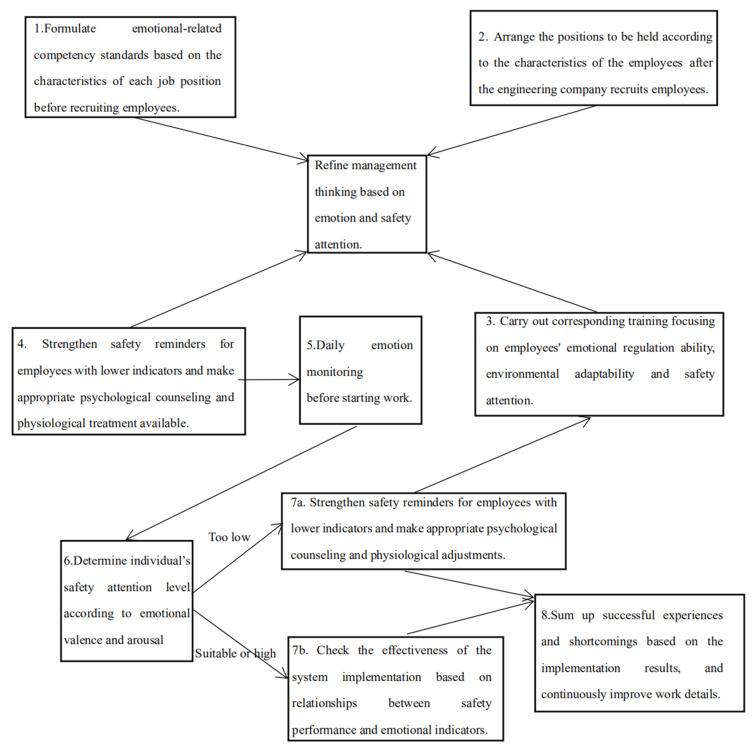
Refined management model based on emotion and safety attention.

**Table 1 ijerph-18-05511-t001:** Descriptive statistics of samples.

Item	Description	Frequency	Percentage
Age	29	18	25.71%
	30	31	44.29%
	31	21	30.00%
Marital status	Married	31	44.29%
	Unmarried	39	55.71%
Area of birth	Northeastern China	11	15.71%
	Northern China	14	20.00%
	Eastern China	12	17.14%
	Southern China	10	14.29%
	Southwestern China	8	11.43%
	Northwestern China	9	12.86%
	Central China	6	8.57%
Highest education level achieved	Elementary school degree and below	0	0.00%
	Junior high school degree	27	38.57%
	Senior high school degree	43	61.43%
	University degree and above	0	0.00%
Department	Construction Department	70	100.00%
Working groups/teams	Team 1	17	24.29%
	Team 2	19	27.14%
	Team 3	16	22.86%
	Team 4	18	25.71%
Work position	Front-line employee	70	100.00%
Date of employment	March 2018	70	100.00%
Sex	Male	70	100.00%

**Table 2 ijerph-18-05511-t002:** Descriptive statistics of variables.

Variable	Level	*n*	Mean	Maximum	Minimum	Standard Deviation
Emotional valence	within	8400	−4.59	0.93	−0.98	0.25
Arousal	within	8400	−4.09	0.84	0.01	0.09
Hazard recognition	within	8400	0.71	1.00	0.00	0.46
Safety attention	within	8400	1093.47	2799.89	0.00	653.80
Risk tolerance	between	70	1.79	2.88	1.00	0.46
Agreeableness	between	70	3.93	6.00	1.83	0.96
Conscientiousness	between	70	3.96	5.33	2.58	0.61
Neuroticism	between	70	5.39	6.67	3.67	0.79
Extroversion	between	70	3.31	5.33	1.50	0.97
Openness	between	70	4.22	6.25	1.67	1.18

**Table 3 ijerph-18-05511-t003:** Results of validity test.

**KMO and Bartlett’s Test**		
The Kaiser–Meyer–Olkin measure of sampling adequacy		0.776
Bartlett’s test of sphericity	Approx. chi-squared	223.108
	Df	36
	Sig.	0.000

**Table 4 ijerph-18-05511-t004:** Results of consistency and variability tests.

Variable	ICC (1)	ICC (2)	Minimum of *r_wg_*	Maximum of *r_wg_*
Safety attention	0.35	0.974155	0.73	0.91
Emotional valence	0.643	0.992131	0.81	0.95
Arousal	0.584	0.989926	0.79	0.87

## Data Availability

We declare that our data are available anytime.

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
