# Peer review of "Re-Thinking the Mediating Role of Emotional Valence and Arousal between Personal Factors and Occupational Safety Attention Levels"

_ijerph, 2021, doi:10.3390/ijerph18115511_

Round 1
Reviewer 1 Report
Overall, the topic area is very much of interest and the quantitative analysis conducted on the data is comprehensive. However, I have concerns about the highly ambitious nature of the study with very small sample size.
In the methods section on page 6 lines 235 to 243 described the respondents. The concerns I have are as follows:
- The number of persons that participated in the study was 70 and they were from a construction project and were of 29-31 years of age, no experience and from the same nationality. While their responses to 120 images, I'm concerned by the potential for low level of variance.
- Information was not provided on "no work experience" means as the participants are of age that some work experience, whether in construction or other industry, would be expected.
- Socioeconomic characteristics are not provided on the participants including education, household composition, area of birth, and so on.
- Lines 251 to 252 of page 6 states that the "research team meticulously designed 120 pictures containing the construction site under the guidance of multiple experts". Who are these experts and what process was used or why 120 pictures - none of this information is provided.
- In the measures section on page 7 lines 275-276 the authors describe the use of "AI technology to recognize the emotional valence and arousal of the participants in the video". The authors need to provide more detail on the AI technology - how, what and so on.
- For personality measures on page 7 lines 285 to 291 - the authors describe the use of questionnaire developed by Tsinghua University. The authors need to provide more information on the validity and reliability of using this instrument and how it had been used in previous studies. In addition, did the respondents suffer from response exhaustion with completion of 60 statements for personality, 8 construction scenarios for risk tolerance and hazard recognition on 120 pictures.
- What was the average length of time that the respondents spend on completing all the tasks? Where they compensated for their time? What incentive was there for participating in the study? How about for the employer? These need to be detailed or addressed in the study.
In section 3.3 Analytical approach:
- this section require tightening and removal of unnecessary background information on well recognized techniques of SEM.
- The Table presentation of table 1 on line 346 page 8 - the number of digits after the decimal place need to be consistent and generally 2 to 3 maximum.
- This section requires the presentation of descriptive statistics on the respondents.
- Statement on lines 347 to 348 on page 9 is confusing. Are the authors saying that they have tested the reliability and validity of the data or that it should be tested? Not quite clear.
- Unnecessary need to present table 2 as a table - unsure what it adds beyond what is in the paragraph on lines 358 to 359 page 9.
Discussion section on page 13 lines 443-447 of the results of the research being forwarded to 3 public occupational safety and public health management experts - I'm unsure what this adds to your results.
The paper throughout makes states of academics or researcher or literature without providing reference to who they are referring (see lines 479-482 on page 13). In addition, they provide names of authors such as on page 7 lines 292 to 293 Gross J.J. and John O.P. in 2003 - need appropriate referencing style and reference to their study.
The paper claims throughout (see page 14 lines 515 to 519) that traditional occupational safety attention research mainly focused on the fields of mining, transportation and sports at the organisational or environmental level. However, a quick google scholar search provided a number of studies in this area and using personality measures that had not been referred to by the authors - I would advise the authors to broaden their search.
Significance of the study on page 15 lines 567 to 579 refers to actions that can be taken by the employers. However, I'm wondering whether they are possible given the usually highly subcontractual, fast paced and domination by small and medium sized enterprises of the construction sector.
The paper needs a thorough grammar and English check and revision. It is quite choppy in the writing style.
Author Response
Dear reviewer,
Thank you for your valuable comments. Your comments are of great value to improving the quality of our articles. We have revised them one by one in accordance with your comments. Allow me to express my sincere respect and gratitude to you on behalf of our research team. In the manuscript, besides the grammar check throughout the article, the fonts marked red are thoroughly revised parts according to the reviewers’ comments.
Terribly sorry, because we were worried about the length of the article before, we did not elaborate on the details of the experiment. We apologize for the confusion we caused to you. We have made a comprehensive and thorough revision in accordance with your comments, and added a detailed description on the experiment.
This research was organized by the Department of Construction Management of Tsinghua University. As the main organizers, we worked hard for nearly a year from applying for ethical review to completing the experiment. Although there were not too many participants, 70 participants and 8400 sets of repetitive experiments were enough to find the relationship between emotion and safety attention level. In addition to the 50,000 RMB compensation for the participants, we also spent nearly 60,000 RMB on the purchase of experimental equipment and so on. The research on emotions and engineering security really fascinates us. We hope to contribute to the safety of construction projects through our own efforts.
Besides, we also wrote the limitations of the samples in Section 6. Owing to the research conditions, the sample in this study mainly came from a certain construction site in China. Therefore, future researchers should select employees from different countries as a study sample, to validate the conclusions of this study further. We sincerely welcome scholars from all over the world to discuss this issue, and we also hope the revised article can be clearer and more valuable.
Point 1: The number of persons that participated in the study was 70 and they were from a construction project and were of 29-31 years of age, no experience and from the same nationality. While their responses to 120 images, I'm concerned by the potential for low level of variance.】
Response 1:
Thank you so much for your doubts. Our research team has conducted a comprehensive and in-depth analysis of your comments.
First, according to descriptive statistical analysis, the variance of the sample is suitable, and there is no high degree of similarity in the data of the samples.
Second, because this article focuses on the impact of risk tolerance, personality, and emotions on safety attention, the selection of samples with the same age, gender, and work experience aims to control the impact of these variables.
Third, this does not mean that the sample is too special and lacks representativeness, because the vast majority of Chinese construction workers are young and middle-aged Han men. In addition, the hometowns of the samples selected in this study are different cities all over China. As a country with a large population and a large territory, there are also huge differences within Chinese. For example, many western scholars who understand China point out that the differences between people in Beijing and Shanghai in language, food, clothing, culture, customs, etc. are not fewer than the differences between French and Germans. Therefore, the samples are still with considerable differences.
In summary, our research team thinks that the selected samples are suitable for research.
Point 2: Information was not provided on "no work experience" means as the participants are of age that some work experience, whether in construction or other industry, would be expected.
Response 2:
Sorry, due to language problems, we were not able to accurately express the specific meaning of "no work experience" before. We apologize for the trouble we caused to you. "No work experience" in this research refers to "no work experience related to the construction industry".
First, the sample selected in this article was the same group of employees recruited by China State Construction Engineering Group Co., Ltd(CSCEC) in the spring of 2018, whose training hours were the same when we made the experiment (end of 2019).
Second, the researchers selected 70 volunteers who had no work experience related to the construction industry previously as samples from hundreds of volunteers who signed up.
Third, the design of this research took the work experience of other industries into account. There were indeed samples of work experience in other industries among the 70 people, but their work experience is about the light industry (handicraft industry, textile industry, etc.) and has nothing to do with the construction industry.
Point 3: Socioeconomic characteristics are not provided on the participants including education, household composition, area of birth, and so on.
Response 3:
We collected a number of socio-economic characteristics of the subjects, mainly including education level, area of birth, marital status and other information. The only problem is that it is difficult for us to obtain the household composition, because from the sight of many Chinese people, this is a private topic. But the marital status can be used as a helpful reference for the household composition. In addition, the occurrence of a safety incident is a momentary matter and this article focuses on the impact of construction workers’ short-term emotional valence and arousal on safe attention, which means that the emotion in this article is an instantaneous or short-term physical indicator. As a result, a person’s socio-economic characteristics may not have a significant or direct impact on emotions in such a short period of time.
Point 4: Lines 251 to 252 of page 6 states that the "research team meticulously designed 120 pictures containing the construction site under the guidance of multiple experts". Who are these experts and what process was used or why 120 pictures - none of this information is provided.
Response 4:
The experts are professors from the Department of Construction Engineering Management and Department of Psychology of Tsinghua University. These experts had rich research experience in the fields of emotion, human factor engineering, and construction engineering safety.
The design of 120 images was a complete process. In the first step, the research team selected 60 common safety hazards under the guidance of experts. In the second step, the research team was divided into six groups. Based on the 60 hidden hazards, we went to six construction sites and took multiple images related to a certain hazard at a construction site with unsafe scenes, and other images related to the same hazard at other construction sites with safe scenes. In the third step, we refined the selection of the images and selected the clearest images of the same hidden danger (one for the safe scene and one for the unsafe scene). In the fourth step, under the guidance of experts, the research team divided each image into an area of interest (AOI) to which the study participants were expected to pay attention for determining whether the construction site is safe. In the fifth step, after the experts’ re-view, we also engaged three workers and three engineers in a pre-experiment. In addition to some detailed questions, the participants indicated that the questions were clear, settings were reasonable, and results were reliable. Finally, we carefully modified the details to make them perfect. In summary, the research team meticulously designed 120 images containing construction sites, under the guidance of multiple experts. Aiming to identify whether there was a potential hazard in each image, the research team identified AOIs to which the study participants were supposed to pay attention. In each image, the construction site was divided into two areas: "safe" (with no potential hazard) and "dangerous" (with potential hazards).
Point 5: Section on page 7 lines 275-276 the authors describe the use of "AI technology to recognize the emotional valence and arousal of the participants in the video". The authors need to provide more detail on the AI technology - how, what and so on.
Response 5:
Sorry, we did not elaborate on the AI technology because of the worry about the limitation of article length. After careful revision, we added enough details on the AI technology.
The research team used the FaceReader analysis system developed by the Dutch Noldus Information Technology Company. The system is a professional software used to analyze facial expressions automatically, which can analyze emotion-related indicators accurately and objectively, including emotional valence and arousal. The perfect operation of its workflow depends on a powerful database, the AAM model, and the deep face model. As a mature analysis system, FaceReader has been used by many scholars. At present, more than 1300 articles have been successfully published after adopting the FaceReader system. Therefore, our analysis system was reliable.
Point 6: For personality measures on page 7 lines 285 to 291 - the authors describe the use of questionnaire developed by Tsinghua University. The authors need to provide more information on the validity and reliability of using this instrument and how it had been used in previous studies.
Response 6:
This questionnaire, developed by the Department of Psychology, Tsinghua University, has been relatively mature, with good results in its application in the past 10 years. The questionnaire had 60 statements, that is, 12 questions were listed in detail for each personality trait. Participants scored their conformity from 1 (completely inaccurate) to 7 (completely accurate) based on the description. After the questionnaire was completed, we chose the average personality trait score to represent the participants’ personalities. At present, many published studies have applied the questionnaire; for example, it was used in the article “Personality factors and safety attitudes predict safety behavior and accidents in elevator workers” published in the “International Journal of Occupational Safety and Ergonomics.” (DOI: 10.1080/10803548.2018.1493259). The validity and reliability of this questionnaire was tested in Section 4.1 (measurement model evaluation).
At present, various research institutions have developed many versions of the Big Five Personality Questionnaire, but many questionnaires are too complicated with redundant parts, which reduces the efficiency and accuracy of the test. Therefore, seeking to ensure the accuracy, reliability, and simplicity of questionnaires, this research team adopted a questionnaire designed by the Department of Psychology of Tsinghua University.
Point 7: In addition, did the respondents suffer from response exhaustion with completion of 60 statements for personality, 8 construction scenarios for risk tolerance and hazard recognition on 120 pictures.
Response 7:
Thank you for reminding us. Because the three experiments were not completed consecutively and the participants had enough rest, we were able to prevent the participants from fatigue. First, the participants finished two questionnaires in the morning, which spent the participants about half an hour. Second, the research team provided the participants with a comfortable lunch break that could replenish sufficient energy, to ensure that they had sufficient physical strength to complete the test in the afternoon. Third, the hazard recognition test lasted 75–90 min. Because the young participants all have at least a junior high school degree, and every test in Chinese middle schools takes at least two hours, the participants could easily complete the test within 90 minutes. Finally, in the return visit after the experiment, participants generally stated that they were in a good state of mind when and after answering the questions, and they did not feel tired or uncomfortable; that is, the experiment did not cause any harm. In summary, the subjects did not suffer from response exhaustion.
Point 8: What was the average length of time that the respondents spend on completing all the tasks? Where they compensated for their time? What incentive was there for participating in the study? How about for the employer? These need to be detailed or addressed in the study.
Response 8:
Sorry we didn’t explain relative issues in detail before. First, the total time of the three experiments was about 2h, which did not take the participants too much time. Second, it is worth emphasizing that the experiment was conducted during the employees' holidays, so as not to delay their normal work. Third, the employer was the Department of Construction Management of Tsinghua University. Fourth, the Department of Construction Management contacted the Infrastructure Department of Tsinghua University, and the Infrastructure Department recruited volunteers from China State Construction Engineering Group Co., Ltd(CSCEC). Fifth, each volunteer received sufficient compensation, including a free lunch, a subsidy of 500 RMB and a gracious souvenir from Tsinghua University. To summary, our research team spent nearly 50,000 RMB to compensate the participants.
Point 9: This section require tightening and removal of unnecessary background information on well recognized techniques of SEM.
Response 9:
Thank you for your valuable suggestions. We have deleted the redundant parts related to the SEM according to your comments.
Point 10: The Table presentation of table 1 on line 346 page 8 - the number of digits after the decimal place need to be consistent and generally 2 to 3 maximum.
Response 10:
We are sorry, after copying and pasting the data from Microsoft Excel to the manuscript, there was a format error, which resulted in inconsistent decimal places. We have modified it according to your comments.
Point 11: This section requires the presentation of descriptive statistics on the respondents.
Response 11:
We have provided descriptive statistical information of the respondents according to your comments, please refer to Table 1.
Point 12: Statement on lines 347 to 348 on page 9 is confusing. Are the authors saying that they have tested the reliability and validity of the data or that it should be tested? Not quite clear.
Response 12:
Sorry, due to language translation issues, we failed to clearly express the original intention, and we apologize for the inconvenience caused to you. Our original intention was that the reliability and validity of the data had been tested, and the following paragraphs were specific descriptions of the process.
Point 13: Unnecessary need to present table 2 as a table - unsure what it adds beyond what is in the paragraph on lines 358 to 359 page 9.
Response 13:
Thank you for your reminder, and we have deleted it according to your comment.
Point 14: Discussion section on page 13 lines 443-447 of the results of the research being forwarded to 3 public occupational safety and public health management experts - I'm unsure what this adds to your results.
Response 14:
Sorry, our original intention was to prove the reliability of the conclusions of this experiment and improve the details of the results. Since you think this may not be suitable, we have deleted it. And we apologize for the inconvenience caused to you.
Point 15: The paper throughout makes states of academics or researcher or literature without providing reference to who they are referring (see lines 479-482 on page 13). In addition, they provide names of authors such as on page 7 lines 292 to 293 Gross J.J. and John O.P. in 2003 - need appropriate referencing style and reference to their study.
Response 15:
Thank you so much for your comments. We have added references and revised the author's name format.
Point 16: The paper claims throughout (see page 14 lines 515 to 519) that traditional occupational safety attention research mainly focused on the fields of mining, transportation and sports at the organizational or environmental level. However, a quick google scholar search provided a number of studies in this area and using personality measures that had not been referred to by the authors - I would advise the authors to broaden their search.
Response 16:
Terribly sorry, our previous literature search was relatively limited, and we apologize for the trouble caused to you. We have modified all similar expressions in the full manuscript to make them consistent with the current research status.
We have explained the personality measurement in detail in the answer to the sixth question. At present, many published studies have applied the questionnaire; for example, it was used in the article “Personality factors and safety attitudes predict safety behavior and accidents in elevator workers” published in the “International Journal of Occupational Safety and Ergonomics.” (DOI: 10.1080/10803548.2018.1493259.)
Point 17: Significance of the study on page 15 lines 567 to 579 refers to actions that can be taken by the employers. However, I'm wondering whether they are possible given the usually highly subcontractual, fast paced and domination by small and medium sized enterprises of the construction sector.
Response 17:
Thank you so much for your comments, and we think this is a very valuable question. Your question prompted us to think deeply and improve the suggestion part.
On the one hand, we have added some suggestions for small and medium-sized enterprises in accordance with your opinions. Possible methods include joint implementation by many small and medium-sized enterprises. Some Chinese companies such as State Grid are trying to establish safety systems related to emotions. In addition, measures such as strengthening training proposed in this article are what small and medium-sized enterprises urgently need. At the same time, this also requires the joint efforts of the entire industry and the public departments. Therefore, this article also makes recommendations from the perspective of the public sector.
On the other hand, the development of science and the popularization of technology generally takes some time. At present, smart construction, smart monitoring and smart buildings are still in a stage of rapid development. Some technologies are not yet mature, the cost of whom is relatively high. It will take a cycle for emerging technologies to spread from large enterprises to small and medium-sized enterprises. However, the analysis above doesn’t mean that this type of management model cannot be adopted in small and medium-sized enterprises at present, because the equipment for identifying emotional valence and arousal is not very expensive and takes very little time. As for the fast pace of work, we are supposed to realize that improving the safety attention will make companies pay less accident compensation and get better social reputation, which will indirectly increase corporate profits. Therefore, although small and medium-sized enterprises have a fast pace of work, this method is achievable and should be applied.
Point 18: The paper needs a thorough grammar and English check and revision. It is quite choppy in the writing style.
Response 18:
Thank you so much for your suggestion. We have carried out a comprehensive grammar check and revision, and then hired a professional editing agency to effectively improve the article to make it meet the publication standards.
We deeply appreciate your consideration of our manuscript, and we look forward to hearing from you. If you have any question, please don’t hesitate to contact us at the address below.
Thank you and best regards.

Reviewer 2 Report
Thank you for the opportunity to read this paper, I enjoyed it. While I will recommend that the manuscript can be accepted, overall, this is a very interesting manuscript. However, it would have been good to cover the implications of the findings for research and practice and for other countries need to be covered. I do not agree that there are few studies in construction industry, page 2 line 47. I am not sure how you arrived at this.
You must have captured sensitive data? How did you ensure the data was collected safely? How do you ensure that the details of the respondents in the data were protected? So you test for the reliability by accessing only one scale? Ethics approval must be presented in detail, it is very important.
The data is interesting and there is a good discussion but some points above could be covered to improve the data. I noted the further research recommendations.
Author Response
Dear reviewer,
Thank you for your valuable comments. Your comments are of great value to improving the quality of our articles. We have revised them one by one in accordance with your comments. Allow me to express my sincere respect and gratitude to you on behalf of our research team. In the manuscript, besides the grammar check throughout the article, the fonts marked red are thoroughly revised parts according to the reviewers’ comments.
Terribly sorry, because we were worried about the length of the article before, we did not elaborate on the details of the experiment. We apologize for the confusion we caused to you. We have made a comprehensive and thorough revision in accordance with your comments, and added a detailed description on the experiment.
This research was organized by the Department of Construction Management of Tsinghua University. As the main organizers, we worked hard for nearly a year from applying for ethical review to completing the experiment. In addition to the 50,000 RMB compensation for the participants, we also spent nearly 60,000 RMB on the purchase of experimental equipment and so on. The research on emotions and engineering security really fascinates us. We hope to contribute to the safety of construction projects through our efforts.
Point 1: However, it would have been good to cover the implications of the findings for research and practice and for other countries need to be covered.】
Response 1:
Thank you for your valuable suggestion, and we think this is a valuable comment. This article has been modified to deepen the significance of these results for research and practice.
First, in terms of research, after this revision, this article further clarified the research gap and its theoretical significance.
Second, in practice, this article continued to improve management methods from the perspective of managers, and added matters that should be paid attention to from the perspective of employees, as well as the future policies of the governments.
Third, we believe that the results based on the study have certain significance for the research and practice of construction engineering safety in other countries, so they can be promoted or checked in other places.
Besides, we also wrote the limitations of the samples in Section 6. Owing to the research conditions, the sample in this study mainly came from a certain construction site in China. Therefore, future researchers should select employees from different countries as a study sample, to validate the conclusions of this study further. However, the development of science is a process from point to surface and from part to whole, and the research from a sample from one country can be verified or doubted by samples from other countries. We sincerely welcome scholars from all over the world to discuss this issue and make contribution to improving the safety level of the construction industry together.
Point 2: I do not agree that there are few studies in construction industry, page 2 line 47. I am not sure how you arrived at this.】
Response 2:
Terribly sorry, our previous literature search was relatively limited, and we apologize for the trouble caused to you. We have modified all similar expressions in the full manuscript to make them consistent with the current research status.
Point 3: You must have captured sensitive data? How did you ensure the data was collected safely? How do you ensure that the details of the respondents in the data were protected? So you test for the reliability by accessing only one scale? Ethics approval must be presented in detail, it is very important.】
Response 3:
Yes, the data obtained in this study had a certain degree of sensitivity. Ensuring data security, ethics, and personal privacy are issues to which we attach great importance. We achieved this goal mainly from the following aspects.
Before data collection, we obtained ethical approval and signed ethical agreements with the study subjects. Then, we communicated with the subjects well and made them fully informed.
During data collection, unrelated personnel were strictly prohibited from entering and taking photos at the test site, to prevent data theft. Researchers could only bring the allowed communication and filming equipment into the test site.
After data collection, we anonymized all the information and disrupted the analysis to protect privacy effectively. Our research team deleted all name-containing data and used labels 1–70 for pronouns. In addition, we did not collect private highly sensitive information, such as mobile phone numbers, credit card information, and various passwords.
After the analysis was completed, the data were stored on a hard drive of a safe laboratory computer, thus effectively preventing the data from being stolen. Moreover, we deleted all privacy-related data from the computer that was used for data analysis and shredded the relevant files to make them unrecoverable.
We deeply appreciate your consideration of our manuscript, and we look forward to hearing from you. If you have any question, please don’t hesitate to contact me at the address below.
Thank you and best regards.

Round 2
Reviewer 1 Report
Dear Authors;
Thank you for re-submitting your paper. I recognize the improvements and adoption of suggestions made from the previous version.
In your method section on page 6 lines 231 to 240 I have the following queries:
- Line 232 "...carried out this research as an employer, to study the mediating effect of emotional valence..." Is the construction site used for this study Tsinghua University?
- Lines 234-235 "..contact a Chinese construction engineering company and recruit hundreds of volunteers into this study" - Can you please be more specific on the number of volunteers as hundreds can mean 200 or 20000.
- Lines 235-236 "After repeated screening, the research team selected the 70 participants" - What was the screening process? Need to provide this in the appendix or detailed description of inclusion and exclusion criteria.
- Descriptive statistics table provided on page 6-7 lines -246-247 - this is generally provided in the results section and not the methods section. As well, in the table for "Education level" - is this the highest education level achieved? if so please retitle it. You have data for Working group/teams and Team 1, Team 2, Team 3 and Team 4 without description or discussion of what this means.
Under experiment preparation lines 250-257 page 7 - this needs to be reworded and statement of the data that was collected from the participants provided and not information on what wasn't. In addition, there is reference to the survey instrument/questionnaire that was completed by the respondents - the questionnaire (or translation of it) should be included or reference to the questions in it be made.
On page 9 lines 316-317 "As a mature analysis system, FaceReader has been used by many scholars" - please provide reference to the studies and how it had been used.
Post visits descriptions on page 9 lines 320 to 322 on the participants being in good state of mind when they answered the questions. What questionnaire was used? Was it the same as the questionnaire prior to the process of examining the images? What was the time period between them?
Page 9 lines 335-336 "After data collection, we anonymized all the information and disrupted the analysis..." - what does disrupted mean here? Please clarify.
Page 9 lines 340 to 343 the paragraph is italic - is there a reason for it?
In the section 3.2 on page 9 line 344 - I would recommend including a short sentence introducing this section.
In the measures on personality page 10 lines 358-360 there is description of the questionnaire being mature and having many applications. In addition, on lines 363-367 "At present, many published studies..." please provide references to these studies and hyperlink to the questionnaire. In addition, description is made on lines 363 "we chose the average personality trait score..." is this how the questionnaire was designed to be used.
Page 11 lines 404 to 409 - description of Mplus8.3 - collapse this. You only need a very brief sentence detailing the software that you had used for your analysis. It appears that you had used both Mplus8.3 and SPSS - please state both and provide the version number.
Table results 2 page 11 lines 414-415 - this is presented without a clear discussion. Please tell us what it means and what we are reading. In addition, you used 3 decimal places for the data - is there a reason for this specificity?
On page 11 lines 514 - 520 discussion - strong statements are made on the safety research and it may suggest that it would be country specific? Please double check as your references are for studies conducted in China and Hong Kong.
In the discussion and elsewhere - care needs to be taken on statements and language used. For example on page 18 lines 651-653 "Large enterprises should place a demonstrative and leading role ...". This is an academic paper and care needs to be taken with possible rewording of this to "Large enterprises are recommended to place a demonstrative and leading role...". Words like "should" are quite strong.
Page 19 lines 704-707 on future research - possibly consider adding other construction sites from different locations. In addition, the placing of statements "We sincerely welcome scholars from all over the world to discuss this issue" - this would not be the location for this and the text for it.
Overall - the paper requires a critical read to check for start of sentences (for example page 16 lines 582 the paragraph in a new section begins with "Because" and tenses. In addition, the references require rechecking as some titles are in capital letters such as reference 17.
My best of luck to the researchers.
Author Response
Dear reviewer,
Thank you so much for your valuable comments. We sincerely admire your rigor and conscientiousness. Not only did we improve the quality of the article, but also deeply learned the rigorous scientific spirit once again. We have revised them one by one in accordance with your comments. Allow me to express my sincere respect and gratitude to you on behalf of our research team.
In the manuscript, besides the grammar check throughout the article, the fonts marked blue are thoroughly revised parts according to your comments. The attachment is proof of editing and polishing.
We are fascinated by the research of emotions and engineering safety, so we conducted experiments and wrote this article. From applying for ethical review to completing the experiment, we worked hard for nearly a year. And we hope to contribute to the safety of construction projects through our own efforts.
Point 1: Line 232 "...carried out this research as an employer, to study the mediating effect of emotional valence..." Is the construction site used for this study Tsinghua University?
Response 1:
Thank you so much for your comments. Our research team took this issue into consideration when we designed the experiment. First, the participants in this experiment were from a construction site located at Tsinghua University and the Infrastructure Department of Tsinghua University helped us contact them. Second, although this study used 120 pictures for the hazard recognition experiment, the construction sites in these photographs were from other construction sites in Beijing and had nothing to do with Tsinghua University. This measure avoided the deviation caused by memory and self-interest when workers were asked to answer questions related to the construction site where they work.
We apologize for the confusion caused to you, and we have added an explanation of this issue in the article. For more details, please refer to line 281-284.
Point 2: Lines 234-235 "...contact a Chinese construction engineering company and recruit hundreds of volunteers into this study" - Can you please be more specific on the number of volunteers as hundreds can mean 200 or 20000.
Response 2:
Thank you for your doubts and we have revised this statement. Because this experiment had strict privacy protection measures, all data related to unselected volunteers were deleted. At present, we aren’t able to restore the data, but according to our memories, the specific number of all volunteers was around 310, and we selected 70 workers to participate in the experiment. We are terribly sorry that we aren’t able to provide an accurate number due to privacy protection, and we have deleted the phrase "hundreds of". For more details, please refer to line 233.
Point 3: Lines 235-236 "After repeated screening, the research team selected the 70 participants" - What was the screening process? Need to provide this in the appendix or detailed description of inclusion and exclusion criteria.
Response 3:
The screening process was divided in four steps. First, we collected basic information including every volunteer’s age, work experience, sex, ethnicity, email address, and health status through questionnaires. In the second step, the research team selected volunteers in good health of the same or similar age (29–31), work experience, sex (male), and ethnicity (Han). In the third step, we asked the selected volunteers to participate in the experiment via email, and expressed our gratitude to the unselected volunteers. Finally, to protect privacy, the research team deleted all the information of the unselected volunteers. For more details, please refer to line 236-242.
Point 4: Descriptive statistics table provided on page 6-7 lines -246-247 - this is generally provided in the results section and not the methods section. As well, in the table for "Education level" - is this the highest education level achieved? if so please retitle it. You have data for Working group/teams and Team 1, Team 2, Team 3 and Team 4 without description or discussion of what this means.
Response 4:
Thank you for your valuable suggestions, and we have moved the table to the results section according to your requirements.
Yes, education level refers to the highest education level achieved. We have retitled it.
Teams 1–4 refer to the four parallel working groups of the project from which we recruited participants. They took turns to finish the work, and their work content was the same.
For more details, please refer to line 430-437.
Point 5: Under experiment preparation lines 250-257 page 7 - this needs to be reworded and statement of the data that was collected from the participants provided and not information on what wasn't. In addition, there is reference to the survey instrument/questionnaire that was completed by the respondents - the questionnaire (or translation of it) should be included or reference to the questions in it be made.
Response 5:
Thank you for your suggestion. We have revised it according to your opinion.
The questionnaire to collect basic information of the participants has 11 questions including age, sex, ethnicity, area of birth, marital status, highest education level achieved, time of employment, department, work position, and work group.
For more details, please refer to line 258-260.
Point 6: On page 9 lines 316-317 "As a mature analysis system, FaceReader has been used by many scholars" - please provide reference to the studies and how it had been used.
Response 6:
Noldus's official website states that FaceReader has been used in 1,300 published papers. We searched “FaceReader” in the web of science or Scopus database, and we indeed found some papers related to FaceReader.
For example, Chia-Yin Y. et al. published the paper “Applying FaceReader to Recognize Consumer Emotions in Graphic Styles” in “Procedia CIRP”. (DOI: 10.1016/j.procir.2017.01.014.)
Shin published “Development of Calculating Formula for Elementary School Students’ Scientific Positive Emotions through FaceReader-Focused on Life Science Videos” in the journal “Biology Education”. (DOI: 10.15717/bioedu.2017.45.2.226).
Hadinejad et al. published “Emotional responses to tourism advertisements: the application of FaceReader” in the journal “Tourism Recreation Research”. (DOI: 10.1080/02508281.2018.1505228.)
According to literature review, the current use of FaceReader by other scholars is similar to ours; that is, using this technology to recognize the emotions of the participants through facial expressions, so as to carry out research in their own fields. Therefore, the research method adopted in this article has sufficient basis.
For more details, please refer to line 327-330.
Point 7: Post visits descriptions on page 9 lines 320 to 322 on the participants being in good state of mind when they answered the questions. What questionnaire was used? Was it the same as the questionnaire prior to the process of examining the images? What was the time period between them?
Response 7:
Thank you so much for your question. The return visit was conducted about two weeks after the experiment. Since the purpose of this return visit was mainly to confirm whether the participants had any adverse reactions after the experiment and whether the compensation was paid, the previous questionnaires are not suitable. There were only three questions in this questionnaire: (1) What is your participant number? (2) Did you have any adverse reactions after the test? If yes, please write down specific symptoms. (3) Have you received compensation and souvenirs from us?
According to the results, all participants stated that they were in a good state of mind during and after the test and they did not feel tired or uncomfortable; the experiment did not cause any harm. Moreover, they all received compensation from our research team, including ¥500 and a souvenir from Tsinghua University. This proved our experiment did not cause any adverse effects on the health of the participants.
For more details, please refer to line 331-338.
Point 8: Page 9 lines 335-336 "After data collection, we anonymized all the information and disrupted the analysis..." - what does disrupted mean here? Please clarify.
Response 8:
We apologize for the misunderstanding caused by improper translation. Our original intention was to anonymize the data and then analyze it to protect privacy. We did not mean "disrupted the analysis". We have deleted this word and changed this sentence to “After data collection, we anonymized all the information and then started the analysis to protect privacy effectively.”
Point 9: Page 9 lines 340 to 343 the paragraph is italic - is there a reason for it?
Response 9:
Terribly sorry. This is due to formatting issues during language editing, and we have changed its format. We sincerely apologize for the inconvenience caused to you.
Point 10: In the section 3.2 on page 9 line 344 - I would recommend including a short sentence introducing this section.
Response 10:
Thank you very much for your suggestion, and it is valuable for improving the expression of our article. We have added a short sentence to introduce this part according to your request:
“To conclude, the variables measured in this study mainly included emotional valence, arousal, personality, risk tolerance, hazard recognition performance, and safety attention. The following are the specific measurement methods for the variables.”
For more details, please refer to line 358-360.
Point 11: In the measures on personality page 10 lines 358-360 there is description of the questionnaire being mature and having many applications. In addition, on lines 363-367 "At present, many published studies..." please provide references to these studies and hyperlink to the questionnaire. In addition, description is made on lines 363 "we chose the average personality trait score..." is this how the questionnaire was designed to be used.
Response 11:
We are sorry that due to language problems, our original intention was not clearly translated into English. Our original intention was that this questionnaire had been used in some published works, not many. The translator exaggerated the scope of application of this questionnaire. We sincerely apologize to you for this problem and have checked the full article to eliminate similar errors. In future article writing, we will also pay attention to this issue.
However, there do exist some published patents and articles that used this questionnaire, whose link is https://www.wjx.cn/jq/42637341.aspx.
To ensure the accuracy, reliability, and simplicity of questionnaires, this research team adopted a questionnaire.
This questionnaire, designed by the Department of Psychology of Tsinghua University, is relatively mature with good results in its application in the past 10 years. The questionnaire has 60 pictures; 12 pictures are listed in detail for each personality trait. The person with full hair in the picture is the “central figure” who exhibits a certain behavior. There are 1–2 prompt words beside each picture to describe the content of the picture. The participants were asked to assume that they were the “central figures” and then assess the possibility that they exhibited the behavior in the picture. Participants scored their possibility from 1 (0%) to 7 (100%) based on the description. After the questionnaire was completed, for every participant, we took the average score of the 12 questions as the final score of this personality trait.
At present, there do exist some published patents and articles that used this questionnaire:
Rau, P.P.; Liao, P.C.; Guo, Z. (2018). Personality factors and safety attitudes predict safety behavior and accidents in elevator workers. International Journal of Occupational Safety & Ergonomics, 2018:1-9. DOI: 10.1080/10803548.2018.1493259.
Wenyu L.; Cheng W.; Xin H. (2020). Quantitative Personality Predictions from a Brief EEG Recording. IEEE Transactions on Affective Computing, DOI: 10.1109/TAFFC.2020.3008775.
Dan Z.; Xin H.; Fei W. (2017). A personality measurement method and device based on brain-computer interface technology. Invention patent, patent number, ZL 2017 1 0405323.2.
Wenyu L.; Xin H.; Xuefei L. (2020). EEG responses to emotional videos can quantitatively predict big-five personality traits, Neurocomputing, in press.
Jingjing C.; Baoshun G.; Yuhang J. (2019). Personality assessment system V1.0 based on EEG technology. Computer software copyright, registration number, 2019SR1071257.
For more details, please refer to line 374-384.
Point 12: Page 11 lines 404 to 409 - description of Mplus8.3 - collapse this. You only need a very brief sentence detailing the software that you had used for your analysis. It appears that you had used both Mplus8.3 and SPSS - please state both and provide the version number.
Response 12:
Thank you for your advice, and we have collapsed the introduction of Mplus8.3 according to your requirements to make the article clear and concise. We are sorry that we didn’t indicate the version of SPSS, and we have added the version number of SPSS (SPSS23.0).
Point 13: Table results 2 page 11 lines 414-415 - this is presented without a clear discussion. Please tell us what it means and what we are reading. In addition, you used 3 decimal places for the data - is there a reason for this specificity?
Response 13:
We are sorry that we didn’t explain this in detail because of concerns about the length of the article. Now we have explained the contents of the table in detail based on your suggestions.
There is no special reason for the 3 decimal places. According to your comment "the number of digits after the decimal place need to be consistent and generally 2 to 3 maximum" in your last review, we chose 3 decimal places. If you think this is inappropriate, we have modified it to 2 decimal places.
Point 14: On page 11 lines 514 - 520 discussion - strong statements are made on the safety research and it may suggest that it would be country specific? Please double check as your references are for studies conducted in China and Hong Kong.
Response 14:
Thank you for your suggestions. In fact, we never wanted to make a strong statement. We are sorry that the inaccurate translation led to this problem, and have modified these statements to make them suitable:
From “Given that the global construction industry continues to face severe safety and health issues” to “Given that the global construction industry continues to face some safety and health issues difficult to solve”.
From “we believe that our present findings have clear and important theoretical and practical significance with respect to improving occupational safety and public health” to “we think that our present findings may have certain theoretical and practical significance with respect to improving occupational safety and public health”.
From “this article provided a novel perspective and method for occupational safety and public health research” to “this article may provide a relatively novel perspective and method for occupational safety and public health research”.
Actually, the statements made in this article do not target specific countries and regions. We believe that the results based on this study have certain significance for the research and practice of construction engineering safety in other countries, so they can be promoted or checked in other places. Besides, we also wrote the limitations of the samples in Section 6. Owing to the research conditions, the sample in this study came from a certain construction site in Beijing, China. Therefore, future researchers may select employees from different places as a study sample, to validate the conclusions of this study further.
However, the development of science is a process from point to surface and from part to whole, and the research based on a sample from one country can be verified or doubted by samples from other countries. We sincerely welcome scholars all over the world to discuss this issue and make contribution to improving the safety level of the construction industry together.
Importantly, after this revision, our references have become more abundant. Among the 54 references in this article, there are 17 references whose authors come from Hong Kong or the mainland of China, and the rest 37 references whose authors are from all over the world, account for about 70%.
Point 15: In the discussion and elsewhere - care needs to be taken on statements and language used. For example, on page 18 lines 651-653 "Large enterprises should place a demonstrative and leading role ...". This is an academic paper and care needs to be taken with possible rewording of this to "Large enterprises are recommended to place a demonstrative and leading role...". Words like "should" are quite strong.
Response 15:
We are so sorry for the inaccurate expressions. Please accept our apology. According to our intention, we didn't want to make the statements strong. After receiving your comments, we have carefully checked the article and revised 17 strong statements to make our expressions appropriate. For example, in many sentences, we changed "should" to "be recommended to", "be expected to", "be supposed to" and other phrases, and then we added sentences such as "we suggest" and "we advise" to ensure our expressions not so strong.
Point 16: Page 19 lines 704-707 on future research - possibly consider adding other construction sites from different locations. In addition, the placing of statements "We sincerely welcome scholars from all over the world to discuss this issue" - this would not be the location for this and the text for it.
Response 16:
Thank you for your advice. After discussion, we agree that your comments are very important to improve our article. We have written your suggestion "possibly consider adding other construction sites from different locations" into this section.
At the same time, we have moved the statement "we sincerely welcome scholars from all over the world to discuss this issue" to the "acknowledge" section.
Point 17: Overall - the paper requires a critical read to check for start of sentences (for example page 16 lines 582 the paragraph in a new section begins with "Because" and tenses. In addition, the references require rechecking as some titles are in capital letters such as reference 17.
Response 17:
Thank you so much for your suggestion. We have carried out a comprehensive grammar check and revision. After last editing, we hired a professional editing agency again to effectively improve the article to make it meet the publication standards. As for the format problem in the reference, we have corrected it in accordance with your comments and journal requirements. The attachment is proof of editing and polishing.
We deeply appreciate your consideration of our manuscript. Allow us to express our sincere respect and gratitude again. If you have any question, do not hesitate to contact us.
Thank you and best regards.
